# Characterization of Volatile Compounds by HS-GC-IMS and Chemical Composition Analysis of Colored Highland Barley Roasted at Different Temperatures

**DOI:** 10.3390/foods11182921

**Published:** 2022-09-19

**Authors:** Cong Wang, Zhiming Zhang, Xiayin Zhang, Xinyi Tian, Kai Chen, Xiaoxiong Zeng

**Affiliations:** 1College of Food Science and Technology, Nanjing Agricultural University, Nanjing 210095, China; 2School of Food Science and Pharmacy, Xinjiang Agriculture University, Urumqi 830052, China

**Keywords:** colored highland barley, roasting temperature, HS-GC-IMS, volatiles fingerprints, chemical composition

## Abstract

Colored highland barley (CHB) is featured with its potential health-promoting benefits. CHB is frequently processed through roasting, which changes its volatile smells, color, and composition. The objective of this work was to establish the volatile fingerprints of CHB that had been roasted at different temperatures using E-nose and headspace-gas-chromatography-ion-mobility spectroscopy (HS-GC-IMS). The findings showed that roasting increased the relative contents of pyrazines, aldehydes, and ketones while decreasing the relative contents of alcohols, esters, and sulfides. Pyrazines were identified as the markers for volatile substances of the roasted CHB (RCHB). The outcomes of the principal component analysis (PCA) and hierarchical clustering analysis (HCA) demonstrated that the volatiles could easily distinguish between raw CHB and RCHB instead of differentiating between CHB roasted at different temperatures. Additionally, after roasting, the color characteristics and CHB constituents underwent changes, and the effect of roasting temperature on these changes differed depending on the cultivar. Protein, free amino acids, and flavonoids appeared to primarily participate in the variations of volatile substances, and the free fluorescence intermediary compounds might involve changes in color parameters and aromas. These findings improved our knowledge of the volatiles in CHB that were roasted under various conditions.

## 1. Introduction

Highland barley (*Hordeum vulgare* L. var. *nudum* Hook. f.), one of the variations of the Gramineae family, is widely cultivated in the Qinghai–Tibet Plateau region of China for centuries [1]. Traditionally, it is consumed as an essential staple food and animal feed for local Tibetans [2]. In the past decades, growing evidence demonstrated that the daily intake of highland barley was associated with decreased risks of chronic disease, and its unique chemical composition also showed potential health-promoting benefits, making it superior to those of regular cereals [3]. It was reported that the content of β-glucan in highland barley ranged from 3.66% to 8.62%, which was significantly higher than that of other wheat crops, suggesting it be beneficial for the regulation of blood lipid and promotion of gut homeostasis [4]. The protein of highland barley was generally higher than those of rice, corn, and wheat, with a range from 8.1% to 20.3%, contributing to the reduced prevalence of cardiovascular disease [5]. Moreover, the lower glycemic index of highland barley starch (39.4–47.5) made it suitable for the consumption of diabetics [4]. Colored highland barley (CHB) is special germplasm of hull-less barley, characterized by its pericarp layer color (white, blue, purple, and black) [6]. Recent studies proved that CHB is more nutritious than the white cultivar, and its high amounts of dietary fiber, anthocyanin, and polyphenols were accountable for its antioxidant, neuroprotective, hypoglycemic, and hypolipidemic ability [7,8]. Thus, accelerating the development of CHB as a functional cereal has become the consensus in recent years.

Roasting is commonly accepted and performed to enhance the accessibility of constituents and extend the shelf-life of highland barley owing to its manageable and non-chemical characteristics [9]. Simultaneously, thermal-induced degradation, oxidation, and chemical reaction (e.g., the Maillard reaction) promote the formation of volatile substances, conferring specific sensory characteristics on the roasted highland barley [10]. The roasting condition usually refers to time and temperature, which leads to different impacts on physical and biochemical differences in the sensory profiles. It has been reported that alterations in the numbers and relative contents of volatile substances of highland barley were affected by its variety and chemical composition during the roasting treatment and primarily dominated by heterocycles, esters, and aldehydes, conferring cocoa-like, roasted, and nutty flavors [11]. With the extension of roasting time, apparent changes in the volatiles of highland barley were observed, varying from the domination of alcohols and esters (fruity, flowery flavors) to heterocycles [12]. Nonetheless, the influence of different roasting temperatures on the volatile profiles of highland barley remain poorly understood, let alone the roasted CHB (RCHB). Despite the contribution to the volatile aromas, roasting-induced generation of the Maillard reaction products (MRPs) also resulted in the alterations of color parameters whilst inevitably presenting a detrimental effect on the nutritional property since some MRPs (e.g., acrylamide, carboxymethyl lysine) had been classified as likely carcinogenic for humans [13,14]. Until now, the MRPs produced during the roasting treatment of CHB are seldom studied, particularly at different temperatures.

Headspace-gas-chromatography-ion-mobility spectroscopy (HS-GC-IMS) is currently proven to be suitable for profiling volatile compounds due to its advantages of low detection limits (ppb), no time-consuming pretreatment, and high accuracy [15]. Consequently, it had been successfully adopted in analyzing the volatiles of raw (rice, barley, and oat) and processed products (wheat, corn) [16,17]. The electronic nose (E-nose) offers a nondestructive alternative method of identifying the overall characteristics of odor substances and can be used to supplement HS-GC-IMS for aroma analysis [18]. Hence, to gain a comprehensive understanding of the volatiles of CHB roasted at different temperatures, a combination of HS-GC-IMS and the E-nose is necessary.

Accordingly, this study aimed to explore the volatile profiles of CHB roasted at different temperatures by the E-nose and HS-GC-IMS, as well as establish their volatiles fingerprints. Additionally, the relationship between the volatile compounds and constituents was further proposed, as well as the effect of the roasting temperature on the color characteristics and chemical composition of CHB.

## 2. Materials and Methods

### 2.1. Materials and Chemicals

Highland barley with white (WH), blue (BU), and black (BK) colors were obtained from Xinning Biotechnology Co., Ltd. (Qinghai, China), and the commercial roasted barley was obtained from Hengai Traditional Chinese Medicine Co., Ltd. (Hebei, China). Thermostable α-amylose and amyloglucosidase were obtained from Solarbio Science and Technology Co., Ltd. (Beijing, China). Test kits for the determination of protein and glucose were purchased from Yuanye Bio-Technology (Shanghai, China) and Jiancheng Bioengineering Institute (Nanjing, China), respectively. All other chemicals and reagents were of analytical grade.

### 2.2. Roasting Treatment of CHB at Different Temperatures

The roasting treatment of CHB was performed by an industrial roasting machine (MSDC-5, MS, Changzhou, China) at temperatures of 180.0 °C, 220.0 °C, and 260.0 °C for 9.0 min, separately. Afterward, the kernels were ground and sieved by a 100.0 mesh sifter to obtain the RCHB powder, named WH-180, WH-220, WH-260 (white), BU-180, BU-220, BU-260 (blue), BK-180, BK-220, and BK-260 (black), respectively.

### 2.3. Color Parameters Analysis of the RCHB

The color parameters, including *L** (brightness), *a** (redness), and *b** (yellowness), of the RCHB were measured by a colorimeter (CR-400, Konica Minolta, Japan). Moreover, its browning index (BI) was calculated as follows:
$$x=\frac{\left(a^{*}+1.75\times L^{*}\right)}{5.645\times L^{*}+a^{*}-3.012\times b^{*}}\;BI=\frac{100\times (x-0.31)}{0.17}$$

### 2.4. Volatile Profiles Analysis of the RCHB

#### 2.4.1. Electronic Nose (E-Nose) Analysis

First, 3.0 g of the RCHB was put in an airtight vial and incubated at 60.0 °C for 30.0 min. Thereafter, a needle connected to the Teflon tube of the E-nose and a balanced needle were injected into the vial before the acquisition of data for 120.0 s by a PEN3 portable electronic nose (Airsense Analytics GmbH, Schwerin, Germany) equipped with a sensor array of 10 kinds of semi-conductor metal oxide sensing elements (Appendix A). The signal responses arranged from 80.0–90.0 s were expressed as G/G0 and used for analyzing the relative content of the volatile aromas from CHB roasted at different temperatures. The commercial roasted barley was regarded as the control.

#### 2.4.2. Headspace-Gas-Chromatography-Ion-Mobility Spectroscopy (HS-GC-IMS) Analysis

The volatiles fingerprints of CHB roasted at different temperatures were established by HS-GC-IMS (FlavourSpec^®^, G.A.S., Dortmund, Germany). Briefly, 2.0 g of the RCHB was put in a 20.0 mL headspace vial with the aluminum cover sealed before incubating at 60.0 °C for 15.0 min with vigorous vibration. Thereafter, 0.5 mL of the headspace was injected into the heated injector (90.0 °C) using a heated syringe (90.0 °C) in splitless mode. The measurement was carried out at the flow rates of 2.0 mL/min (0–2.0 min), 2.0–20.0 mL/min (2.0–10.0 min), 20.0–100.0 mL/min (10.0–20.0 min), and 100.0–150.0 mL/min (20.0–30.0 min), respectively, using nitrogen (99.999% of purity) as the carrier gas. The analytes were eluted and separated at 40.0 °C, followed by ionization in the IMS ionization chamber. The retention index (RI) of each compound was calculated using n-ketones C4–C9, and the volatile compounds were identified by comparing RI and drift time (DT) via the GC-IMS library. Finally, the qualitative analysis of each sample was executed through the Gallery plot plug-in of LAV software. The commercial roasted barley was regarded as the control.

### 2.5. Chemical Composition Analysis of CHB Roasted at Different Temperatures

#### 2.5.1. Total Starch, β-Glucan, and Protein Analysis

The content of total starch was determined by the procedure of AACC 76-13.01. The calcofluor fluorescence method was used to determine the content of β-glucan with β-glucan from oat as the standard [19]. The content of free amino acids was determined by the Ninhydrin method with l-leucine as the standard [20]. The protein content was determined according to the instruction of the commercial kit with bovine serum albumin as the standard. Briefly, 20.0 μL of protein extracts, prepared by the alkaline-acid method [21], was incubated with 200.0 μL of the biuret reagent for 15.0 min at an ambient temperature before their absorbances were recorded at 540.0 nm.

#### 2.5.2. Total Polyphenols Analysis

The content of total polyphenols was analyzed based on a previous report with slight modifications [22]. Briefly, 1.0 g of the RCHB powder was dispersed in 30.0 mL of methanol (95%) before sonicating at an ambient temperature for 30.0 min. After centrifuging at 4000.0× *g* for 5.0 min, the supernatant was collected, and the sediment was mixed with new methanol to start the secondary extraction. After repeating for three cycles, the gathered supernatants were combined and used for the determination of total polyphenols and flavonoids. Specifically, 50.0 μL of the extract was incubated with 125.0 μL of the Folin–Ciocalteu reagent for 10.0 min in a dark atmosphere. Afterward, 80.0 μL of a sodium carbonate solution (150.0 mg/mL) was added and incubated for an additional 30.0 min before its absorbance was recorded at 760.0 nm. The content of total polyphenols was expressed as g gallic acid equivalents/100.0 g of the RCHB.

#### 2.5.3. Total Flavonoids Analysis

The content of total flavonoids was determined based on a previous study with minor modifications [23]. Briefly, 100.0 μL of the extracts was mixed with 30.0 μL of a sodium nitrite solution (5.0%) and incubated at ambient temperature for 5.0 min. Thereafter, 30.0 μL of aluminum chloride solution (5.0%) was added and incubated for another 6.0 min. Finally, 400.0 μL of a sodium hydroxide solution (4.0%) was added to the mixture, which was kept in a dark atmosphere for an additional 20.0 min before its absorbance was recorded at 510.0 nm. The content of total flavonoids was expressed as g rutin equivalents/100.0 g of the RCHB.

#### 2.5.4. Free Fluorescence Intermediary Compounds Analysis

The formation of free fluorescence intermediary compounds (FIC) was analyzed by suspending the RCHB powder in 6.0% of SDS solution at a ratio of 1:50, followed by stirring at ambient temperature for 30.0 min [24]. Afterward, the slurry was filtrated and the FIC in the filtrates was determined by a microplate reader (Varioskan Flash, Thermo Electron Corporation, Vantaa, Finland) with the wavelength of excitation at 370.0 nm and emission at 440.0 nm, respectively. The FIC value was achieved by comparing the fluorescence intensity of samples with that of 0.05 μg/mL of quinine sulphate in a form of the mean value versus arbitrary fluorescence units (FU). Quinine sulphate dissolved in 0.2 M of sulphuric acid was employed as the fluorescence standard, and 0.05 μg/mL of quinine sulphate revealed a fluorescence emission of 100.0 arbitraries units [25].

### 2.6. Statistical Analysis

The results were given as means ± standard deviations and the statistical analysis was conducted by one-way analysis of variance (ANOVA) with Duncan’s multiple range tests. The value of *p* < 0.05 was considered statistically significant. Statistical analyses were primarily performed using SPSS 23.0 software (SPSS Inc., Chicago, IL, USA), and the principal component analysis (PCA) was conducted using SIMCA 14.0 software (Umetrics, Umea, Sweden).

## 3. Results and Discussion

### 3.1. Color Parameters of CHB Roasted at Different Temperatures

Color is one of the significant properties of roasted products and an indicator of general quality control owing to the browning and caramelization reactions during processing [26]. The degree of roasting can also be reflected by the alteration of the color parameters under certain circumstances [27]. In the case of barley malt, its roasting degree could be evaluated according to the development of high-molecular-weight browning compounds [28]. The association between roasting degree and color parameters is also well-established in coffee and hazelnuts [29,30]. Therefore, the color parameters of the RCHB were essential to be analyzed. As revealed in Table 1, the visual color of CHB tended to turn dark brown with roasting temperature regardless of the color differences of the cultivars. The *L** value of the white cultivar was found to be lower after roasting compared to that of WH and to vary adversely with roasting temperature, which was consistent with other studies that showed the *L** values of barley and highland barley flour were inversely proportional to roasting temperature [31]. At the roasting temperature of 260.0 °C, the *L** value of only the blue cultivar seemed to considerably drop (*p* < 0.05), while the *L** values of BU-180 and BU-220 did not statistically differ from those of BU. The *L** values of the black cultivar and its roasted counterparts were noticeably lower than those of the white and blue cultivars whilst the elevated roasting temperature had a limited impact on its *L** value, as evidenced by the slight differences between BK, BK-180, BK-220, and BK-260. In terms of *a** values, significant increases followed by decreases were found in the white and blue cultivars with the elevated roasting temperature (*p* < 0.05), in which temperatures of 180.0 °C and 220.0 °C promoted their values and a temperature of 260.0 °C presented a contrary impact. Despite that, the *a** value of BU-260 was also obviously increased by 48.74% compared to that of BU (*p* < 0.05). In contrast, the *a** value of the black cultivar seemed to not be affected at the roasting temperatures of 180.0 °C and 220.0 °C, whereas a remarkable decrease was observed at the roasting temperature of 260.0 °C as its *a** value (BK-260) decreased by 17.66% (*p* < 0.05). It appeared that the average *b** values of CHB generally increased after roasting, apart from WH-220, which showed a minor decrease and a significantly lower value compared with WH and WH-260. The *b** value of the blue cultivar was significantly promoted at the roasting temperature of 220.0 °C and slightly influenced by other roasting temperatures. Meanwhile, the *b** value of the black cultivar appeared to have an approximate variation with its *L** value during the roasting treatment. Roasting-induced increases in *a** and *b** values had been reported in the cases of highland barley flour, wheat, and sesame seeds, suggesting that the enhancement of redness and yellowness was primarily attributed to the formation of brown pigments derived from the Maillard reaction [1,32,33]. Moreover, the *b** value was revealed to be less affected by roasting temperature, consistent with the findings presented in Table 1. BI usually indicates the purity of the brown color [34]. As expected, the BI values of CHB increased with the roasting temperature within the cultivars whilst the BI value of WH-260 was apparently higher than that of BU-260 and BK-260 (*p* < 0.05). Overall, these results suggested that the color parameters of the blue and black cultivars were less affected by the roasting temperature due to their relatively higher contents of anthocyanin and colored polyphenols [35].

### 3.2. Volatile Compounds Analysis of CHB Roasted at Different Temperatures

#### 3.2.1. Volatile Aromas Analyzed by E-Nose

The volatile aromas of the RCHB attributed to the responses of E-nose sensors are shown in Appendix A, which reveals that the aromas sensitive to W1C, W3C, W5C, W1W, W2W, and W1S were apparently changed. Further hierarchical clustering analysis (HCA) indicated that the volatile aromas were generally divided into three clusters (Figure 1a), including cluster 1 (W1C, W3C, W5C, and W1S), cluster 2 (W2S, W3S, W5S, and W6S), and cluster 3 (W1W and W2W), respectively. Similarly, CHB and its roasted counterparts were also classified into three groups, in which the raw CHB was assigned to one group and was well discriminated from the RCHB and roasted barley. Besides, WH-180, WH-220, BU-180, BU-220, BK-180, and BK-220 were gathered and appeared to be distinctly separated from the group composed of WH-260, BU-260, and BK-260. Preliminary PCA results revealed that the first two principal components accounted for over 96.40% of the total variance (80.10% for PC1 and 16.30% for PC2) in the volatile aromas (Figure 1b). W1W and W2W were observed to be positively related to WH-260, BU-260, BK-260, and the roasted barley, suggesting relatively higher responses to aromatic compounds and sulfur organic compounds due to their low sensory detection thresholds whilst remaining flavor active (Appendix A), consistent with the previous findings in the case of the roasted barley [36]. WH-180, WH-220, BU-180, BK-180, and BK-220 were found to be associated with W1S, W1C, and W3C, apparently distinct from BU-220, which showed high relevance to W5C. Similarly, the raw CHB was discriminated from the RCHB and presented a close association with W2S, W3S, W5S, and W6S. These findings showed that roasting CHB significantly changed its volatile aromas, which varied depending on roasting temperature and were not cultivar-dependent.

#### 3.2.2. Volatiles Fingerprints Established by HS-GC-IMS

The HS-GC-IMS analysis was carried out, and the volatiles fingerprints were also established, to further show the qualitative and relatively quantitative information of the RCHB’s volatile profiles. As revealed in the topographic plots (Appendix A), most of the signals appeared and were well separated within the retention time of 100.0–800.0 s and the drift time of 1.0–1.5 ms. Consistent with the observations of the E-nose, the roasting treatment of CHB apparently promoted the numbers and concentrations of its volatile compounds, as evidenced by the enhanced signal intensity of the spots, in which the red one indicated a higher concentration of the substance than the raw CHB, whereas the blue one indicated a lower concentration [37]. After comparing the topographic plots with the GC × IMS library, a total of 55.0 compounds were identified including 49.0 monomers and 6.0 dimer derivates (Table 2). A further gallery plot revealed that 36.0 compounds were altered after roasting including 11.0 aldehydes, 7.0 alcohols, 4.0 esters, 4.0 pyrazines, 4.0 ketones, 4.0 acids, and 1.0 furan, in which furfural, 5-methylfurfural, 3-methylthiopropanal, 1-heptanol, 2-acetylpyrazine, methylpyrazine, trimethylpyrazine, 1-hydroxy-2-propanone, n-propyl acetate, butanoic acid, acetic acid, 2-methylbutanoic acid, and 1,4-dioxane were newly generated regardless of 3.0 dimer derivates (Figure 2a). Generally, the relative contents of furfural, 5-methylfurfural, 3-methylthiopropanal, 1-heptanol, 1-hydroxy-2-propanone, n-propyl acetate, and acetic acid were promoted with roasting temperature, which impacted the contents of 2-acetylpyrazine, trimethylpyrazine, butanoic acid, and 1,4-dioxane varied with the cultivars. On the contrary, methylpyrazine and 2-methylbutanoic acid were negatively affected by roasting temperature. Besides, 10.0 compounds including hexanal, propanal, nonanal, heptanal, 1-propanol, pentan-1-ol, (Z)-3-hexen-1-ol, methyl 2-methylpropanoate, benzene acetaldehyde, acetic acid, and hexyl ester also revealed noticeable decreases in their contents with roasting temperature, in which (Z)-3-hexen-1-ol and benzene acetaldehyde were not detected at the roasting temperature of 260.0 °C. On the other hand, the alteration of some compounds was largely dependent on the varieties during the roasting treatment. It was observed that 2-butanone, 2-methylpropanal, and 2-acetylfuran appeared to be the novel volatiles of the black cultivar, whereas 2-butanone and (Z)-3-nonen-1-ol were newly discovered in the white and blue cultivars after roasting, respectively. Additionally, (Z)-3-nonen-1-ol in the black cultivar could not be detected after roasting, suggesting a thermolabile property.

HCA results showed that the raw CHB and RCHB were assigned to three groups, in which group 1 (WH, BU, and BK) was well discriminated from RCHB and roasted barley (Figure 2b). Similarly, WH-260, BU-260, and BK-260 were also gathered with the roasted barley (group 3), featured by their higher intensities of furfural, 1-heptanol, 4-methylpentanol, and acetone. For the RCHB clustered in group 2, 3-methylbutanal, methylpyrazine, and ethyl butyrate appeared to be their representative volatile compounds. Besides, the raw CHB was primarily characterized by nonanal, propanal, and ethyl butyrate. Further PCA results revealed that the raw CHB and its counterparts roasted at the temperature of 260.0 °C were separately clustered and well differentiated from CHB roasted at the temperatures of 180.0 °C and 220.0 °C on the basis of volatiles with the first two principal components accounting for over 78.50% of the total variance (58.80% for PC1 and 19.70% for PC2, Figure 2c). Despite the alterations in the numbers and concentrations of individual compounds, the relative contents of volatile substances including sulfides, pyrazines, furans, ketones, esters, acids, aldehydes, and alcohols were noteworthy to be analyzed. As revealed in Figure 2d, the contents of alcohols, esters, and sulfides of CHB were decreased after roasting, in which esters and sulfides were reduced with roasting temperature. Contrarily, the contents of aldehydes, ketones, acids, furans, and pyrazines were generally elevated after roasting, and the impact of roasting temperature on their contents varied with the cultivars. It appeared that aldehydes were the primary volatile substances attributed to CHB and the RCHB as evidenced by their largest proportions in the total volatiles.

Meanwhile, the highest promotion in the contents of volatile substances was observed in pyrazines with their values increased at the average rates of 2589.58% (white), 2514.58% (blue), and 3004.30% (black), respectively. Pyrazines were thus recognized as the characteristic aroma indicators for the RCHB. It was still noteworthy that the relative contents of furans and acids in the RCHB were negatively associated with their roasting temperature, and the highest elevations in furans and acids among CHB roasted at 260.0 °C were observed in BK-260 and WH-260, revealing 43.16% and 447.44% higher contents than those of BK and WH, respectively. Under the same circumstances, the highest amounts of furans and acids appeared to be BU-260 (1.83%) and BK-260 (4.80%), separately, indicating that more acids were generated than furans, contrary to the findings in the roasted coffee bean [38].

Pyrazines and aldehydes were usually demonstrated as the primary volatile substances of the roasted products (e.g., cocoa, soybean) and derived from the α-aminoketones by Strecker degradation and lipid oxidation, conferring nutty (furfural), earthy (trimethylpyrazine), cocoa (methylpyrazine), and roasted flavors (5-methylfurfural) [17,39,40]. As reported, the roasting temperature was a crucial factor that influenced the formation of pyrazines and aldehydes, presenting a positive and negative correlation with their contents [38], separately. Notably, such correlation was also observed in the RCHB (Figure 2d). Ketones were another key oxygenated component that originated from the Strecker degradation of alcohol oxidation or/and ester decomposition, conferring pungent, sweet-caramel (1-hydroxy-2-propanone), and fragrant flavors (2-butanone) [41,42,43]. Their interactions with aldehydes were essential for the formation of pyrazines, particularly under high roasting temperatures [42]. Alcohols were normally recognized as the products of the oxidation degradation of fatty acids and presented fewer contributions to the aroma characteristic due to their relatively high threshold, whereas esters were described as the second most important volatiles after pyrazines during the roasting treatment, conferring favorable fruity flavors [41,44]. Recent literature had suggested that alcohols and esters were sensitive to the roasting treatment, and the high temperature promoted their loss through chemical degradation or/and volatilization [45], consistent with the findings of the current study (Table 2). Nonetheless, there were still some alcohols and esters that arose during the roasting treatment such as 1-propanol and n-propyl acetate, conferring sweaty and fruity flavors [39], respectively. Volatile acids were generally reduced during the roasting treatment, whereas the Maillard reaction induced the formation of several acids such as butanoic acid, acetic acid, and 2-methylbutanoic acid, which presented sour, fruity, and vinegar-like flavors [44,46]. Simultaneously, furans and sulfides were reported to be generated via the pyrolysis reaction and conferred crucial influences on the flavors due to their low odor threshold, despite their low proportions in the volatile substances [47]. In summary, these results indicated that the characteristic volatile fingerprints of the RCHB could be established through HS-GC-IMS and the non-targeted characteristic compounds were not suitable for the differentiation of CHB roasted at different temperatures.

### 3.3. Chemical Composition of CHB Roasted at Different Temperatures

The chemical constituents of the RCHB were assessed and summarized in Table 3. As revealed, the contents of total starch in the white and blue cultivars were significantly promoted at the roasting temperatures of 260.0 °C and 180.0 °C with their values increased by 21.95% (WH-260) and 17.41% (BU-180) compared with those of WH and BU (*p* < 0.05), respectively. In contrast, no obvious differences were observed among their counterparts roasted at other temperatures within the cultivars. The total starch content of the black cultivar appeared to be slightly influenced by roasting temperature, exhibiting a non-statistical difference from its roasted counterparts. Accordingly, these results suggested that the alteration in the starch amount of CHB was not directly dependent on roasting temperature, consistent with the previous findings in the cases of roasted waxy-barley and hull-less barley [13,48]. Nonetheless, it was still noteworthy that the roasting treatment of highland barley resulted in dramatic changes in the morphology, structure, and functional property of its starch, and such variations had already been well elucidated in the literature [3]. Generally, the β-glucan content of CHB was increased after roasting, followed by a decrease in roasting temperature. It appeared that WH-180, BU-180, and BK-180 revealed the highest contents of β-glucan within their cultivars (*p* < 0.05), and their values were increased by 20.19%, 39.39%, and 20.53% compared with those of WH, BU, and BK, respectively. Similar increases in the content of β-glucan had been reported in the case of roasted oats, likely attributed to its deconstruction with the non-cellulosic polysaccharides induced by roasting. Instead, the impact of roasting at the temperature of 260.0 °C on the β-glucan content of CHB varied with the cultivars, in which WH-260 and BK-260 showed −17.63% and 12.90% higher values than WH and BK (*p* < 0.05), whereas a statistically equal level was observed between BU-260 and BU. Roasting resulted in an apparent decrease in the protein content of CHB with the reduction rates ranging from 2.72% to 44.22% (white), 10.38% to 60.53% (blue), and 44.24% to 58.99% (black), in agreement with the findings in the cases of roasted highland barley and maize [12,49]. Simultaneously, the content of free amino acids of CHB also appeared to be reduced with roasting temperature, which might be ascribed to its participation in the Maillard reaction, as evidenced by the alteration in the BI values and generations of volatiles (Table 1 and Table 2) [50].

The impact of roasting temperature on the contents of total polyphenols and flavonoids of CHB varied with the cultivars. Specifically, the roasting temperatures of 180.0 °C and 220.0 °C had slight influences on the contents of total polyphenols of the white cultivar. In contrast, the phenolic compounds of the blue cultivar were significantly decreased at the roasting temperature of 180.0 °C (*p* < 0.05), suggesting their sensitivity to the thermal-induced decomposition (e.g., anthocyanin, cumaric acid, and quercetin), in accordance with the previous findings [35,51,52]. Notably, the polyphenols contents in the white and blue cultivars were obviously promoted at the roasting temperature of 260.0 °C, which might be associated with the formation of polyphenol derivatives such as the degradation of caffeic acid, which induced the generation of phenylindans, and the oxidation of polyphenols, which induced the generation of melanoidins during the thermal processing [53]. The MRPs might also contribute to the changes in their polyphenol contents [54]. Besides, roasting had little impact on the polyphenols content of the black cultivar (*p* > 0.05). The total content of flavonoids of CHB decreased with roasting temperature, and its value was significantly reduced by 23.03–68.48% (white), 12.75–65.10% (blue), and 37.01–66.23% (black) (*p* < 0.05), as supported by the findings in the case of roasted barley [9].

Furthermore, the Maillard reaction induced the formation of FIC during the roasting treatment of CHB and is analyzed and revealed in Figure 3 and Table 4. It was found that their fluorescence spectrum appeared to exhibit one characteristic peak with a wide range of wavelengths (Figure 3), consistent with the observation of the previous report [55]. As expected, the roasting treatment of CHB resulted in an apparent increase in its FIC value whilst presenting a roasting temperature-dependent manner with rates of increase of 53.72–968.37% (white), 12.63–718.32% (blue), and 33.94–454.10% (black) (Table 4), respectively. Consequently, the FIC values among CHB roasted at 260.0 °C were significantly different from each other (*p* < 0.05), in which WH-260 (1.15) showed the highest value followed by BU-260 (0.92). Roasting-induced promotions in FIC values had been observed in buckwheat and groats, where FIC values of the roasted buckwheat showed relevance to the roasting time [14,56]. It was noteworthy that the FIC values of the RCHB were apparently lower compared with other referred products, which might be ascribed to the differences in the determination wavelength. It was suggested that the FIC were a heterogeneous group of compounds with characteristic pigmentation, and their generation usually affected the color parameters [24]. Evidently, similar alterations in BI and FIC values were observed in the RCHB. Besides, it should be noted that the FIC values of the RCHB were obviously lower than that of the roasted barley (1.48 ± 0.001) (*p* < 0.05), suggesting a potentially higher roasting degree than the RCHB [27]. Nonetheless, the FIC values appeared to be the indicators for MRPs and were unable to elucidate the specific compounds, which requires further information obtained by chromatography and mass spectrometry analyses.

### 3.4. Correlation between the Chemical Composition and Volatile Substances of CHB Roasted at Different Temperatures

Since volatiles typically resulted from physical and chemical reactions involving chemical ingredients during roasting, their link needed to be clarified. Free amino acids were favorably correlated with esters and sulfides and negatively correlated with ketones and pyrazines, as shown in Figure 4 (*p* < 0.05), indicating that their involvement in the Maillard reaction was essential for the production of volatile components of the RCHB [41]. As a result of its thermally induced breakdown and interactions with other aroma precursors, the protein was found to have a considerably negative connection with pyrazines, furans, and acids and a significantly positive correlation with alcohols (*p* < 0.05) [36,38]. Polyphenols and flavonoids showed opposite associations with esters (negative and positive, respectively), whereas flavonoids also showed negative relevance to pyrazines, ketones, and acids (*p* < 0.05), consistent with the previous reports, which demonstrated that several phenol acids (e.g., gallic acid and caffeic acid) and naringenin might be involved in suppressing the formation of Maillard-type products [57,58]. Furthermore, it was found that flavonoids were positively correlated with both protein and free amino acids (*p* < 0.05), indicating that they interacted in a specific way to produce volatile compounds during the roasting process [59].

## 4. Conclusions

A total of 55.0 compounds were found in the current investigation, of which 36.0 compounds were evidently altered and 13.0 compounds were newly formed. The volatile fingerprints of CHB roasted at different temperatures were established. Further investigation showed that pyrazines dramatically increased with roasting temperature, demonstrating the volatile indications that were typical of the RCHB. The findings of HCA clustering and PCA showed that the non-targeted characteristic chemicals were not acceptable for discriminating CHB roasted at various degrees but could easily distinguish between raw CHB and RCHB. The roasting treatment also resulted in the alteration in the constituents contents of CHB, in which protein, free amino acids, and flavonoids participated in the changes of pyrazines, furans, ketones, esters, and acids (*p* < 0.05). In addition, the effect of roasting temperature on color characteristics, volatile compounds, and chemical contents differed from the cultivars of CHB, necessitating more research.

## Figures and Tables

**Figure 1 foods-11-02921-f001:**
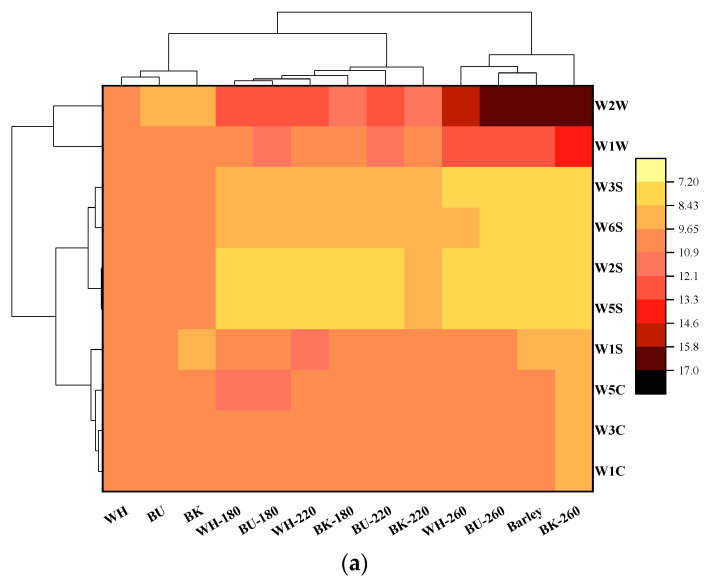
Heatmap clustering (**a**) and biplot of PCA (**b**) of the volatile aromas attributed by E-nose in the raw and CHB roasted at different temperatures.

**Figure 2 foods-11-02921-f002:**
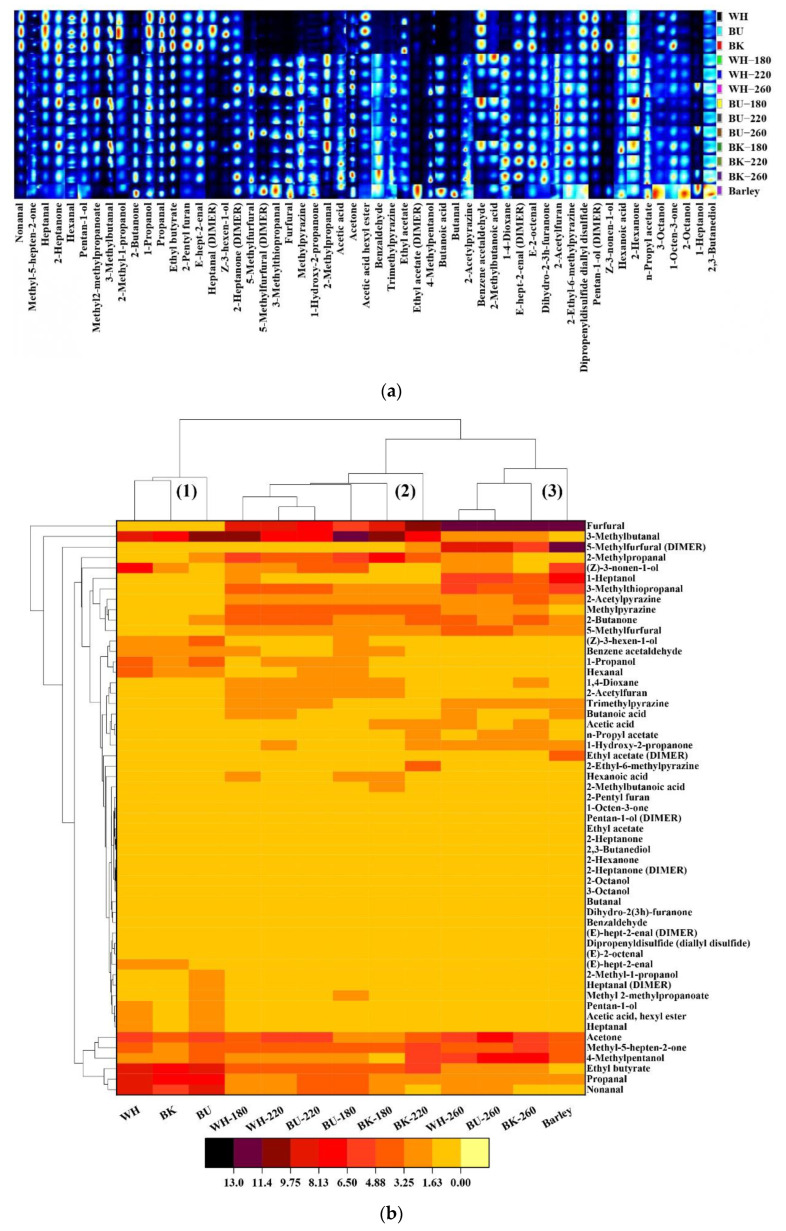
Gallery plot (**a**), heatmap clustering (**b**), and biplot of PCA (**c**) of the volatile compounds attributed by HS-GC-IMS and the relative contents of volatile substances (**d**) of the raw and CHB roasted at different temperatures. The numbers in the biplot were consistent with that of the identifiers listed in Table 2.

**Figure 3 foods-11-02921-f003:**
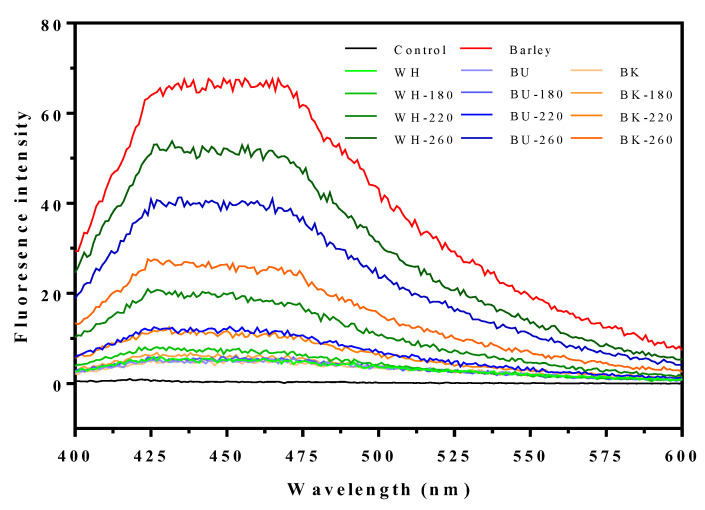
Fluorescence spectrum of the FIC in the raw and CHB roasted at different temperatures.

**Figure 4 foods-11-02921-f004:**
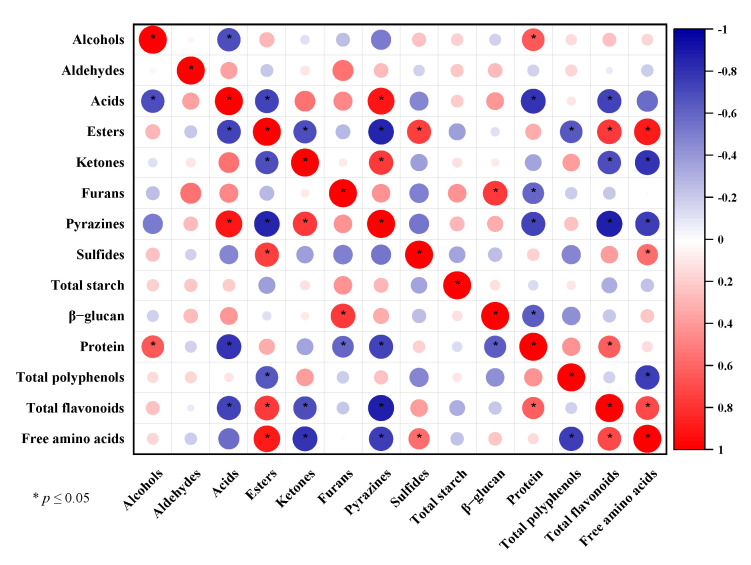
Correlation plot of the chemical composition and the volatile substances of the raw and CHB roasted at different temperatures.

**Table 1 foods-11-02921-t001:** The color parameters of CHB roasted at different temperatures.

Samples	*L**	*a**	*b**	BI
WH	56.79 ± 1.44 ^a^	7.85 ± 0.68 ^cd^	26.53 ± 1.16 ^ab^	69.58 ± 0.77 ^ef^
WH-180	53.00 ± 2.60 ^ab^	9.79 ± 0.92 ^ab^	26.62 ± 1.26 ^ab^	81.37 ± 5.06 ^bc^
WH-220	47.86 ± 2.44 ^bc^	9.73 ± 0.20 ^ab^	24.85 ± 1.50 ^bc^	85.88 ± 3.49 ^b^
WH-260	46.07 ± 1.32 ^c^	7.32 ± 1.32 ^de^	28.11 ± 1.28 ^a^	100.78 ± 2.62 ^a^
BU	52.08 ± 2.13 ^ab^	4.37 ± 0.57 ^f^	21.14 ± 1.23 ^d^	57.11 ± 4.38 ^g^
BU-180	53.51 ± 5.14 ^a^	7.12 ± 0.42 ^de^	23.94 ± 2.12 ^bcd^	67.74 ± 1.46 ^ef^
BU-220	52.83 ± 1.39 ^ab^	10.16 ± 0.10 ^a^	25.50 ± 0.20 ^abc^	78.33 ± 2.10 ^cd^
BU-260	45.80 ± 0.39 ^c^	6.50 ± 0.27 ^e^	22.86 ± 0.41 ^cd^	77.49 ± 2.88 ^cd^
BK	35.81 ± 1.52 ^d^	10.53 ± 0.51 ^a^	14.54 ± 1.09 ^ef^	72.56 ± 2.25 ^de^
BK-180	35.56 ± 5.50 ^d^	10.29 ± 0.81 ^a^	15.89 ± 3.95 ^e^	78.58 ± 6.05 ^cd^
BK-220	36.58 ± 4.55 ^d^	10.14 ± 0.98 ^a^	16.91 ± 2.58 ^e^	81.07 ± 2.05 ^bc^
BK-260	34.43 ± 1.60 ^d^	8.67 ± 0.47 ^bc^	16.63 ± 1.45 ^e^	82.63 ± 3.67 ^bc^

BI: Browning index; Mean values with different letters within the same column differ significantly (*p* ˂ 0.05).

**Table 2 foods-11-02921-t002:** Qualitative and quantitative analyses of volatile compounds in CHB roasted at different temperatures.

NO.	Compound	RI	Dt [RIPrel]	Relative Content (%)												
				WH	WH-180	WH-220	WH-260	BU	BU-180	BU-220	BU-260	BK	BK-180	BK-220	BK-260	Barley
	**Aldehydes**															
1	Nonanal	1096	1.4755	8.5	3.1	2.9	2.4	8.4	3.9	3.4	2.2	5.9	2.6	1.6	1.2	0.54
2	Heptanal	902.5	1.3295	1.9	0.71	0.55	0.35	2.1	0.96	0.62	0.35	1.3	0.75	0.48	0.31	0.2
3	Hexanal	799.8	1.2591	3.4	1.6	1.6	1.1	3.2	1.9	1.8	1.1	2.4	1.6	1.4	1.1	0.54
4	3-Methylbutanal	651.4	1.4028	9.4	10.0	7.1	3	11.0	12.0	7.9	3.1	7.0	11.0	6.5	2.2	1.5
5	Propanal	491.7	1.0481	9.1	3.1	3.2	2.5	8.1	4.1	4.0	2.6	7.0	3.1	3.2	2.5	1.7
6	(E)-hept-2-enal	965.3	1.2613	2.1	0.67	0.91	0.47	1.2	0.56	0.74	0.54	2.0	1.2	1.2	0.75	0.2
7	5-Methylfurfural	970.8	1.1241	nd	2.3	2.4	4	nd	1.8	2.7	4.2	nd	1.9	2.2	3.1	2.2
8	3-Methylthiopropanal	911.3	1.395	nd	4.3	4.5	5	nd	2.5	4.5	4.4	nd	2.6	2.9	3.3	6.2
9	Furfural	827.2	1.3386	nd	8.2	9.6	13	nd	5.7	8.1	13.0	nd	9.1	10.0	12.0	12.3
10	2-Methylpropanal	558.1	1.2883	1.3	6.2	4.3	2	2.4	6.2	4.7	2.4	nd	7.1	3.7	1.4	0.5
11	Benzaldehyde	963.5	1.1501	nd	0.72	0.72	0.68	nd	0.61	0.65	0.61	nd	0.96	1.0	1.1	0.73
12	Butanal	595.5	1.2897	0.53	0.54	0.57	0.53	0.52	0.54	0.53	0.42	0.4	0.55	0.57	0.63	1.5
13	Benzene acetaldehyde	1040.7	1.2566	2.9	2.2	1.0	nd	3.0	2.7	1.2	nd	2.1	2.1	0.78	nd	0.15
14	(E)-2-octenal	1056.4	1.3389	1.2	0.53	0.79	0.43	1.1	0.49	0.72	0.63	1.2	0.8	1.1	0.49	0.24
15	Heptanal (DIMER)	900.9	1.701	1.2	0.32	0.18	nd	2.3	0.46	0.17	0.1	0.81	0.41	0.2	0.12	0.083
16	5-Methylfurfural (DIMER)	968.1	1.4766	nd	nd	nd	9.6	nd	nd	nd	9.6	nd	nd	2.1	5.2	12.5
17	(E)-hept-2-enal (DIMER)	962.2	1.6661	0.86	0.3	0.47	0.42	0.39	0.2	0.3	0.45	1.2	1.1	1.2	1.1	0.18
	**Alcohols**															
18	Pentan-1-ol	760.4	1.2556	2.1	0.8	0.75	0.49	2.0	1.0	0.8	0.48	1.6	0.94	0.77	0.47	0.54
19	2-Methyl-1-propanol	619.9	1.1731	1.4	0.61	0.47	0.45	1.9	0.77	0.51	0.4	0.53	0.74	0.7	0.43	0.33
20	1-Propanol	567.6	1.1142	3.8	1.6	1.8	1.3	3.9	2.3	2.1	1.3	2.1	1.4	1.4	1.0	1.0
21	(Z)-3-hexen-1-ol	847.2	1.5176	3.0	1.1	0.55	nd	4.2	2.0	0.71	nd	2.2	0.92	0.55	nd	0.37
22	4-Methylpentanol	854.3	1.3099	3.1	2.8	3.2	6.4	3.7	1.8	2.8	7.0	2.0	nd	5.0	8.0	4.8
23	(Z)-3-nonen-1-ol	1153.9	1.8295	8.0	2.3	2.5	2.9	nd	4.7	3.7	2.3	3.0	nd	nd	nd	6.3
24	3-Octanol	994.8	1.4076	0.41	0.29	0.23	0.15	0.56	0.3	0.22	0.14	0.3	0.32	0.22	0.17	0.39
25	2-Octanol	993.8	1.4381	0.15	0.23	0.27	0.16	0.24	0.22	0.17	0.12	0.14	0.33	0.38	0.34	0.32
26	1-Heptanol	971.7	1.3899	nd	1.7	1.4	6.0	nd	1.6	1.3	5.6	nd	nd	1.5	3.8	7.7
27	2,3-Butanediol	792.4	1.3617	0.6	0.46	0.47	0.36	0.53	0.46	0.47	0.35	0.5	0.49	0.46	0.5	nd
28	Pentan-1-ol (DIMER)	757.1	1.5174	0.77	0.26	0.19	0.1	0.75	0.31	0.22	0.12	0.61	0.39	0.33	0.12	0.095
	**Ketones**															
29	Methyl-5-hepten-2-one	683	1.4291	3.9	3.9	4.3	3.6	4	3.7	3.9	3.4	3.1	4.4	5.3	5.5	4.3
30	2-Heptanone	888.5	1.2642	0.58	0.47	0.45	0.29	0.57	0.55	0.38	0.26	0.55	0.49	0.35	0.34	0.097
31	2-Butanone	592.1	1.2444	nd	3.5	3.9	3.3	1.8	3.0	3.7	2.8	nd	3.0	3.3	3.9	2.8
32	1-Hydroxy-2-propanone	675.4	1.2253	nd	1.2	1.9	2.4	nd	0.76	1.4	2.2	nd	1.3	1.7	2.9	2.7
33	Acetone	499.6	1.1201	6.0	3.8	6.3	5.4	5.5	3.0	6.1	6.8	3.6	2.5	4.1	6.3	4.1
34	Dihydro-2(3h)-furanone	914.7	1.3031	0.42	0.94	0.93	0.55	0.82	0.49	0.85	0.64	0.4	0.71	1.3	1.0	0.22
35	2-Hexanone	791.2	1.5027	0.15	0.28	0.24	0.17	0.2	0.34	0.26	0.18	0.16	0.28	0.23	0.14	0.13
36	1-Octen-3-one	988.5	1.274	0.55	0.21	0.27	0.16	0.37	0.12	0.18	0.16	0.66	0.41	0.35	0.32	0.17
37	2-Heptanone (DIMER)	889.5	1.6298	0.23	0.37	0.32	0.33	0.3	0.3	0.18	0.2	0.21	0.39	0.26	0.36	0.26
	**Esters**															
38	Methyl 2-methylpropanoate	683	1.4291	1.5	1.0	0.76	0.35	2.1	2.0	0.87	0.45	1.5	1.4	0.86	0.51	0.16
39	Ethyl butyrate	794.2	1.5588	9.1	4.0	4.4	2.2	8.6	4.7	4.4	2.9	7.6	4.6	5.0	2.9	1.2
40	Acetic acid, hexyl ester	1012.2	1.4061	2.3	0.83	0.65	0.51	2.3	0.93	0.74	0.51	1.3	0.6	0.52	0.4	0.23
41	Ethyl acetate	606.7	1.0971	0.78	0.37	0.5	0.36	0.54	0.44	0.52	0.36	0.79	0.41	0.42	0.43	0.43
42	n-Propyl acetate	710.9	1.1573	nd	1.3	1.6	1.6	nd	1.2	1.4	1.7	nd	1.6	1.9	1.8	1.2
43	Ethyl acetate (DIMER)	605.6	1.3403	0.66	0.36	0.34	0.31	0.68	0.4	0.31	0.24	0.79	0.44	0.39	0.37	4.3
	**Acids**															
44	Acetic acid	600.1	1.1547	nd	1.5	1.5	1.7	nd	1.1	1.0	1.4	nd	1.7	1.9	2.1	1.2
45	Butanoic acid	816.2	1.3903	nd	1.9	1.7	1.7	nd	1.2	1.5	1.5	nd	0.46	1.6	1.5	2.0
46	2-Methylbutanoic acid	848.4	1.4718	nd	0.7	0.48	0.21	nd	0.54	0.41	0.19	nd	1.7	0.33	0.2	0.21
47	Hexanoic acid	997.3	1.3012	0.78	2.0	1.5	0.66	1.2	1.8	1.2	0.59	1.2	2.4	1.5	1.0	0.85
	**Pyrazines**															
48	Methylpyrazine	828.8	1.0823	nd	3.5	3.7	2.7	nd	3.4	3.7	2.9	nd	3.5	3.3	2.7	1.4
49	Trimethylpyrazine	1008.5	1.162	nd	1.8	2.4	2.3	nd	1.6	2.8	2.3	nd	1.6	nd	2.8	1.8
50	2-Acetylpyrazine	1023.9	1.1468	nd	3.0	2.9	2.4	nd	2.3	2.7	2.1	nd	3.2	2.3	4.7	3.0
51	2-Ethyl-6-methylpyrazine	1005.1	1.2003	0.32	0.3	0.38	0.44	0.32	0.34	0.53	0.43	0.31	0.53	3.9	0.34	0.23
	**Furans**															
52	2-Pentyl furan	1001.4	1.2561	0.74	0.47	0.35	0.21	0.91	0.67	0.29	0.23	0.95	0.63	0.51	0.16	0.097
53	2-Acetylfuran	910.3	1.1174	0.72	1.9	2.1	1.6	0.56	2.4	2.6	1.6	nd	1.8	0.31	1.2	0.85
	**Sulfides**															
54	Dipropenyldisulfide (diallyl disulfide)	1083.2	1.2123	1.4	0.59	0.62	0.47	1.1	0.7	0.66	0.4	0.84	0.56	1.47	0.37	0.3
55	1,4-Dioxane	702.5	1.3301	nd	2.5	2.6	1.5	nd	2.6	2.9	1.4	nd	2.6	0.52	2.4	nd

RI: Retention time; Dt: Drift time; nd: Not detected.

**Table 3 foods-11-02921-t003:** Chemical composition of CHB roasted at different temperatures.

Samples	Content (%)					
	Total Starch	β-Glucan	Protein	Total Polyphenols	Total Flavonoids	Free Amino Acids
WH	50.38 ± 1.79 ^c^	4.31 ± 0.099 ^de^	8.82 ± 0.96 ^a^	0.063 ± 0.001 ^e^	0.59 ± 0.007 ^bc^	1.65 ± 0.04 ^a^
WH-180	54.53 ± 1.47 ^bc^	5.18 ± 0.40 ^a^	5.11 ± 0.32 ^bc^	0.064 ± 0.002 ^de^	0.54 ± 0.003 ^def^	1.27 ± 0.06 ^c^
WH-220	53.11 ± 2.45 ^bc^	4.88 ± 0.063 ^b^	4.92 ± 0.42 ^bc^	0.069 ± 0.005 ^cde^	0.54 ± 0.010 ^def^	0.98 ± 0.11 ^d^
WH-260	61.44 ± 2.52 ^a^	3.55 ± 0.14 ^g^	8.58 ± 0.24 ^a^	0.095 ± 0.004 ^a^	0.55 ± 0.012 ^de^	0.52 ± 0.01 ^f^
BU	54.62 ± 2.69 ^bc^	3.36 ± 0.059 ^gh^	10.11 ± 1.09 ^a^	0.064 ± 0.002 ^d^	0.64 ± 0.020 ^a^	1.49 ± 0.11 ^b^
BU-180	64.13 ± 3.71 ^a^	4.65 ± 0.18 ^bc^	3.99 ± 0.77 ^c^	0.056 ± 0.001 ^f^	0.52 ± 0.018 ^ef^	1.30 ± 0.05 ^c^
BU-220	56.02 ± 4.02 ^b^	4.58 ± 0.086 ^cd^	4.86 ± 0.42 ^bc^	0.069 ± 0.001 ^cde^	0.55 ± 0.033 ^de^	1.02 ± 0.02 ^d^
BU-260	52.81 ± 0.51 ^bc^	3.52 ± 0.14 ^gh^	9.06 ± 1.53 ^a^	0.086 ± 0.005 ^b^	0.53 ± 0.010 ^def^	0.52 ± 0.02 ^f^
BK	52.03 ± 2.83 ^bc^	3.41 ± 0.39 ^gh^	9.90 ± 0.48 ^a^	0.068 ± 0.001 ^cd^	0.61 ± 0.023 ^ab^	1.54 ± 0.06 ^b^
BK-180	50.68 ± 1.50 ^c^	4.11 ± 0.20 ^ef^	4.06 ± 0.71 ^c^	0.070 ± 0.002 ^cd^	0.56 ± 0.007 ^cd^	0.97 ± 0.02 ^d^
BK-220	52.74 ± 1.96 ^bc^	3.23 ± 0.13 ^h^	5.37 ± 0.31 ^b^	0.068 ± 0.002 ^cde^	0.51 ± 0.003 ^fg^	0.84 ± 0.02 ^e^
BK-260	56.28 ± 1.49 ^b^	3.85 ± 0.067 ^f^	5.52 ± 0.30 ^bc^	0.073 ± 0.005 ^c^	0.48 ± 0.016 ^g^	0.52 ± 0.01 ^f^

Values with different letters within the same column differ significantly (*p* ˂ 0.05).

**Table 4 foods-11-02921-t004:** FIC values in the raw and CHB roasted at different temperatures.

Samples	FIC Values (FU)			
	0.0 °C	180.0 °C	220.0 °C	260.0 °C
White cultivar	0.11 ± 0.002 ^j^	0.16 ± 0.006 ^g^	0.44 ± 0.005 ^d^	1.15 ± 0.007 ^a^
Blue cultivar	0.11 ± 0.004 ^j^	0.13 ± 0.004 ^i^	0.27 ± 0.009 ^e^	0.92 ± 0.003 ^b^
Black cultivar	0.11 ± 0.002 ^j^	0.14 ± 0.004 ^h^	0.26 ± 0.013 ^f^	0.59 ± 0.001 ^c^

Values with different letters differ significantly (*p* ˂ 0.05).

## Data Availability

Data are contained within the article or Appendix A.

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
