# Peer review of "Characterization of Volatile Compounds by HS-GC-IMS and Chemical Composition Analysis of Colored Highland Barley Roasted at Different Temperatures"

_foods, 2022, doi:10.3390/foods11182921_

Round 1

Reviewer 1 Report

The Introduction must be revised a bit by adding a few more recent work and findings on the topic and by making a comparison of the present work with those studies in terms of efficacy and outcomes.

Table 2 It is advised to classify the components into different chemical classes in the table.

Figure 2 They are not readable. Please enhance the figure quality and enlarge the fonts.

Please highlight the major limitations of the study.

Following articles are advised to cite to improve the manuscript quality:

- https://doi.org/10.7506/spkx1002-6630-20180522-312

- https://doi.org/10.7506/spkx1002-6630-20180522-312

- https://doi.org/10.1515/chem-2021-0039

English language needs a major and thorough revision.

The conclusion must be improved and must include take-home-message from the results obtained.

Author Response

Thanks a lot for the comments or advice. The list of changes is as follows:

  1. The Introduction must be revised a bit by adding a few more recent work and findings on the topic and by making a comparison of the present work with those studies in terms of efficacy and outcomes.

Changes: Thanks very much for the suggestions. The recent progress on the volatiles of the roasted highland barley has been updated as follows:

“The roasting condition usually refers to time and temperature, which leads to different impacts on physical and biochemical differences in the sensory profiles. It had been reported that the alterations in the numbers and relative contents of volatile substances of highland barley were affected by its variety and chemical composition during the roasting treatment and primarily dominated by heterocycles, esters, and aldehydes, conferring cocoa-like, roasted, and nutty flavors [11]. With the extension of roasting time, apparent changes in the volatiles of highland barley were observed, varying from the domination of alcohols and esters (fruity, flowery flavors) to heterocycles [12]. Nonetheless, the influence of different roasting temperatures on the volatile profiles of highland barley remain poorly understood, let alone the roasted CHB (RCHB).” Please check the revised manuscript.

  1. Table 2 It is advised to classify the components into different chemical classes in the table.

Changes: Thanks very much for the suggestions. The volatile components have been classified according to their chemical classes. Please check Table 2 in the revised manuscript.

  1. Figure 2 They are not readable. Please enhance the figure quality and enlarge the fonts.

Changes: Thanks very much for the suggestions. The quality and fonts of Figure 2 have been updated. Please check the revised manuscript.

  1. Please highlight the major limitations of the study.

Changes: Thanks very much for the suggestions. Traditionally, headspace-solid phase microextraction gas chromatography-mass spectrometry (HS-SPME-GC-MS) is extensively applied to investigate volatile compounds due to its safety, accuracy, and superiority of mass spectrometry in the identification of substances (doi.org/10.1016/j.foodchem.2015.09.097). However, the vacuum necessary for MS and the difficulties in distinguishing the isomeric compounds from analytes limit the usage of GC-MS, particularly for the complex food matrix (doi.org/10.1016/j.foodchem.2019.126158). Headspace-gas-chromatography-ion-mobility spectroscopy (HS-GC-IMS) combines the high separation capacity of GC and the fast response of IMS, providing a low detection limit and excellent identification of isomers (doi.org/10.1016/j.foodres.2020.109339). Similarly, GC-IMS had its own disadvantage, in that IMS as a GC detector cannot be used for the precise quantification analysis due to the limitation of the GC-IMS library (doi.org/10.1016/j.foodchem.2021.130055). Consequently, the combination of HS-SPME-GC-MS and HS-GC-IMS is preferred in providing a comprehensive understanding of volatile compounds in many cases (doi.org/10.1111/jfbc.13875). Accordingly, the current study can be improved in the following aspects:

(1) As mentioned above, the limitation of the HS-GC-IMS library makes it hard to fully identify all the volatile compounds despite their good separation during the analysis. Therefore, HS-SPMF-GC-MS can be considered as a supplement for volatiles profiling in further study.

(2) The volatiles generated during roasting are derived from the physical and chemical reactions with the involvement of the chemical constituents. In the current study, the overall contents of constituents in the roasted colored highland barley (CHB) were primarily determined by spectrophotometric measurements and some methods may have deficiencies in specificity. For instance, the Folin-Ciocalteu method has been widely used for the determination of total polyphenols content, whereas some Maillard reaction products may also react with the Folin-Ciocalteu reagent, affecting the accuracy of the results (doi.org/10.1021/jf030723c). Although the correlation between the chemical constituents and volatile substances of the roasted CHB has been proposed, the specific ingredients (e.g. glucose and amino acids) involved in the formation of volatile substances and the Maillard reaction products (e.g. polyphenol derivatives) are still necessary to be elucidated in further study.

(3) HCA results of volatile compounds showed that colored highland barley roasted at 180.0 ℃ and 220.0 ℃ were clustered and well discriminated from its counterparts roasted at 260.0 ℃, suggesting that the steep alterations of volatiles occurred during the variation from 220.0 ℃ to 260.0 ℃ (Figure 2b). Therefore, a smaller gradient of roasting temperature (e.g. 20.0 ℃) should better be considered to reveal the gradual changes of volatile substances with roasting temperature.

  1. Following articles are advised to cite to improve the manuscript quality: - https://doi.org/10.7506/spkx1002-6630-20180522-312; - https://doi.org/10.7506/spkx1002-6630-20180522-312; - https://doi.org/10.1515/chem-2021-0039.

Changes: Thanks very much for the suggestions. The articles advised are significant for improving the manuscript and thus cited accordingly. Please check Ref. 11, 12, and 44 in the revised manuscript.

  1. English language needs a major and thorough revision.

Changes: Thanks very much for the suggestions. We have carefully checked and polished the manuscript accordingly. Please check the revised manuscript.

  1. The conclusion must be improved and must include take-home-message from the results obtained.

Changes: Thanks very much for the suggestions. The conclusion of the manuscript has been updated as follows:

“A total of 55.0 compounds were found in the current investigation, of which 36.0 compounds were evidently altered and 13.0 compounds were newly formed. The volatile fingerprints of CHB roasted at different temperatures were established. Further investigation showed that pyrazines dramatically increased with roasting temperature, demonstrating the volatile indications that were typical of the RCHB. The findings of HCA clustering and PCA had shown that the non-targeted characteristic chemicals were not acceptable for the discriminating of CHB roasted at various degrees but could easily distinguish between raw CHB and RCHB. The roasting treatment also resulted in the alteration in the constituents contents of CHB, in which protein, free amino acids, and flavonoids participated in the changes of pyrazines, furans, ketones, esters, and acids (p < 0.05). In addition, the effect of roasting temperature on color characteristics, volatile compounds, and chemical contents differed with the cultivars of CHB, necessitating more research.” Please check the revised manuscript.

Reviewer 2 Report

Some further comments and recommendations for amendments can be found below:

Introduction:

L34-37:also showed potential health-promoting benefits, making it superior to those of regular cereals” Please specify and add more details, and examples.

L: 59-60: “Until now, the MRPs produced during the roasting treatment are seldomly studied, particularly  at different temperatures.” I suggest to consider some examples connected with roasting process: doi.org/10.1021/jf4054287; doi.org/10.1016/j.lwt.2021.111718; doi.org/10.1111/1750-3841.12161; 10.1021/jf500549t; doi.org/10.1021/jf802250j; 10.1016/j.foodchem.2015.09.064

In accordance with previous research:

Zhang, W.G., Zhang, Y., Yang, X.J., Dang, B., Zhang, J., Du, Y., Chen, D.S. GC-MS analysis of volatile flavor substances in different varieties of roasted hulless barley. Food Sci. 2019, 40, 192–201. (In Chinese) 486 11.

Zhang, Y., Zhang, W.G., Dang, B., Yang, X.J., Chen, D.S., Hao, J. Effects of stir-frying time and method on volatile flavor compounds in highland barley. Food Sci. 2020, 41, 271–277. (In Chinese)

which is a novelty in this work? Please, explain.

Material and methods

“The protein content was determined according to the instruction of the commercial kit with bovine serum albumin as  the standard.” I think more details are necessary. Can the authors explain the need to use that technique?  Commonly used is total nitrogen content determined by the Kjeldahl method according to ISO 20483 / Cereals and Pulses-Determination of the Nitrogen Content and Calculation of the Crude Protein Content-Kjeldahl Method; International Organization for Standardization: Geneva, Switzerland, 2013/

Please, explain what is the reason for the determination of TPC . It is well-known that TPC is considered a non-specific method.

Whether the authors have considered FIC data expressed as mean values in the fluorescence intensity (FI)/g sample?

 Results and discussion

L 174-175: “The degree of roasting can also be reflected by the alteration of the color parameters under certain circumstances” – In relation to potentially harmful Maillard reaction product it is so interesting, so please specify and add more details, and examples.

 Tables 1 and 3 - Please, add letters as a superscript. 

L 207-210: “these results suggested that the color parameters of CHB  roasted at different temperatures varied with the cultivars, in which the blue and black cultivars were less affected due to their relatively higher contents of anthocyanin and colored polyphenols in episperm initially”

Please, describe this phenomenon briefly.

L264: “On the contrary, methylpyrazine and 2-methylbutanoic acid  were negatively affected by roasting temperature”  and L292-293: “Contrarily, the contents of aldehydes, ketones, acids, furans, and pyrazines were generally elevated after roasting and the impact of roasting temperature  on their contents varied with the cultivars” L319-320: “As reported, the roasting temperature was a crucial factor  that influenced the formations of pyrazines and aldehydes, presenting a positive and a negative correlation with their contents [36], separately”

Can the author try to explain the reason for this exception?

L 303: “Nonetheless, it was still noteworthy to emphasize that the relative contents of furans and acids decreased with roasting temperature despite their elevations compared with those of the raw CHB.”
 It is worth describing this phenomenon in more detail.

L315-318: “Pyrazines and aldehydes were usually demonstrated as the primary volatile substances of the roasted products (e.g. cocoa, soybean) and derived from the α-amino-  ketones by the Strecker degradation, the Maillard reaction, and lipid oxidation, conferring nutty (furfural), earthy (trimethylpyrazine), cocoa (methylpyrazine), and roasted flavors  (5-methylfurfural)” It should be mentioned earlier - > L294-303 (aldehydes and pyrazines). Moreover, Strecker degradation is one of the most important reactions leading to final aroma compounds in the Maillard reaction.

L382-386: “In contrast, the phenolic compounds of the blue cultivar were significantly decreased at the roasting temperature of 180.0 (p < 0.05), suggesting their sensitivity to thermal-induced oxidation and degradation [49]. Notably, the polyphenols contents in the white  and blue cultivars were obviously promoted at the roasting temperature of 260.0 , which  might be associated with the formation of polyphenol derivatives (e.g. phenylindans) during the thermal processing [50].”

It has been mentioned too briefly. More specific data and results should be presented (on the basis of the previous studies).

L387: “Besides, roasting seemed to have little impact on the polyphenols content of the black cultivar”  It should be emphasized, whether there are statistically significant differences. However, the TPC method is not precise enough to find statistically significant differences in this case.

L392: I suggest using “free intermediate compounds” instead advanced MRPs.

Fig. 3 b - It would be easier and more clearly compare the results in a table.

L400: “It was suggested that the  advanced MRPs were a heterogeneous group of compounds with characteristic pigmentation and their generation usually affected the color parameters.”

Heat treatment during the roasting process affects the formation of potentially harmful Maillard reaction compounds, such as acrylamide, hydroxymethylfurfural,  furfural, and especially advanced glycation end products (AGE) e.g., Nε-(carboxymethyl)lysine). In my opinion, determining the level of a  well-known specific compound in the advanced Maillard reaction stage is a more specific and thus better solution than estimating the number of free intermediate compounds (FIC) characterized by fluorescence.

L: 395- 397: “As expected, the roasting treatment of CHB resulted in an apparent increase in its FIC value whilst presenting a roasting temperature-dependent manner with the raise rates of 53.72 %~968.37 % (white), 12.63 %~718.32 % (blue)..)

Please, discuss the results with other sources and add more information about FIC levels after heat treatment of other raw materials/ products.

 L 423: “whereas flavonoids were also negatively relevant to pyrazines, ketones, and acids (p < 0.05).”

According to previous data, some phenolic compounds suppressed Maillard-type volatiles and carcinogenic product -  acrylamide (doi.org/10.1016/j.foodchem.2019.125008; doi.org/10.1021/tx9001644). Please take this aspect into account in the discussion.

Author Response

Thanks a lot for the comments or advice. The list of changes is as follows:

  1. L34-37: “also showed potential health-promoting benefits, making it superior to those of regular cereals” Please specify and add more details, and examples.

Changes: Thanks very much for the suggestions. The detailed information about the potential health-promoting benefits of black highland barley has been updated as follows:

“It was reported that the content of β-glucan in highland barley ranged from 3.66 % to 8.62 %, which was significantly higher than that of other wheat crops, suggesting it be beneficial for the regulation of blood lipid and promotion of gut homeostasis [4]. The protein of highland barley was generally higher than those of rice, corn, and wheat, with a range from 8.1 % to 20.3 %, contributing to the reduced prevalence of cardiovascular disease [5]. Moreover, the lower glycemic index of highland barley starch (39.4-47.5) made it suitable for the consumption of diabetics [4].” Please check Line 37-43 in the revised manuscript.

  1. L: 59-60: “Until now, the MRPs produced during the roasting treatment are seldomly studied, particularly at different temperatures.” I suggest to consider some examples connected with roasting process: doi.org/10.1021/jf4054287; doi.org/10.1016/j.lwt.2021.111718; doi.org/10.1111/1750-3841.12161; 10.1021/jf500549t; doi.org/10.1021/jf802250j; 10.1016/j.foodchem.2015.09.064.

Changes: Thanks very much for the suggestions. We are apologizing for the misunderstanding caused by our description of the MRPs in the roasted products. The sentence has been updated to “Until now, the MRPs produced during the roasting treatment of CHB are seldom studied, particularly at different temperatures.” Meanwhile, the suggested articles that are beneficial for the improvement of the manuscript have been considered accordingly. Please check Line 68-69 in the revised manuscript.

  1. In accordance with previous research: Zhang, W.G., Zhang, Y., Yang, X.J., Dang, B., Zhang, J., Du, Y., Chen, D.S. GC-MS analysis of volatile flavor substances in different varieties of roasted hulless barley. Food Sci. 2019, 40, 192–201. (In Chinese) 486 11.; Zhang, Y., Zhang, W.G., Dang, B., Yang, X.J., Chen, D.S., Hao, J. Effects of stir-frying time and method on volatile flavor compounds in highland barley. Food Sci. 2020, 41, 271–277. (In Chinese). which is a novelty in this work? Please, explain.

Changes: Thanks very much for the suggestions. As demonstrated in the manuscript, the volatiles of the roasted highland barley had been reported in the previous study. However, the reported investigations of the roasted highland barley were mainly focused on the differences in aromas among different cultivars (under the same roasting parameter, 5.0 min, 105.0 ℃) or at different roasting times (under the same cultivar, 2.0 min, 5.0 min, 11.0 min, and 13.0 min, 105.0 ℃). The differences in the volatile profiles of colored highland barley (CHB) roasted at different temperatures remained unclear. According to the results of the current study, significant alterations in the volatile substances occur at the roasting temperature over 220.0 ℃ (9.0 min), and our preliminary experiment also supported that the roasting temperature and time selected in the reports appeared to result in an inadequate roasting degree of the current cultivars. Besides, the impact of roasting time on the volatiles of CHB was found to be apparently different from those of roasting temperature based on our study (In submitting). Consequently, this study can be regarded as a supplement and continuation of previous reports.

  1. Material and methods:“The protein content was determined according to the instruction of the commercial kit with bovine serum albumin as the standard.” I think more details are necessary. Can the authors explain the need to use that technique? Commonly used is total nitrogen content determined by the Kjeldahl method according to ISO 20483 / Cereals and Pulses-Determination of the Nitrogen Content and Calculation of the Crude Protein Content-Kjeldahl Method; International Organization for Standardization: Geneva, Switzerland, 2013. Please, explain what is the reason for the determination of TPC. It is well-known that TPC is considered a non-specific method.

Changes: Thanks very much for the suggestions. (1) Detailed information for the determination of protein has been updated in the method, please check Line 139-141 in the revised manuscript.

(2) As you suggested, the Kjeldahl Method is the golden standard for the determination of protein. In the current study, our strategy is to prepare the protein extracts first based on our previous study (10.1016/j.ijbiomac.2019.06.145), followed by the determination by biuret reagent due to the consideration of the availability of equipment. In fact, the attempts of using this strategy in investigating the protein content have been performed in the cases of maize and chickpea (10.1016/j.ultsonch.2015.09.007; 10.1016/j.foodhyd.2021.107351), which provide references for us to choose the colorimetric method as an alternative in the beginning. Nonetheless, according to the results of the current study, the protein of CHB is an undeniable participant in the formation of volatile substances and its accurate determination is necessary. Therefore, a profound investigation of the alterations in protein and individual amino acids has been included in our future scheme.

(3) The Folin-Ciocalteu method has been extensively used for the determination of total polyphenols content despite its deficiency in specificity (doi.org/10.3390/foods11030287; 10.1016/j.jcs.2020.103152; 10.1016/j.foodchem.2015.08.083). Normally, roasting-induced degradation of thermolabile polyphenols and generation of the Millard reaction products occur simultaneously, resulting in the alterations of color, volatiles, and the potential functional property. According to the previous report (doi.org/10.1021/jf030723c), some Maillard reaction products may react with the Folin-Ciocalteu reagent, affecting the final results. Consequently, the determination of total polyphenols by the Folin-Ciocalteu method can be considered as supplementary evidence for the changes in color parameters and formations of the Maillard reaction products. A profound investigation of the alterations of free, bound polyphenols as well as the Maillard reaction products will be performed in our future study.

  1. Whether the authors have considered FIC data expressed as mean values in the fluorescence intensity (FI)/g sample?

Changes: Thanks very much for the suggestions. According to the previous report (10.1016/j.jcs.2007.08.012), the free fluorescence intermediary compounds (FIC) data has been defined by comparing the fluorescence intensity of samples with that of 0.05 μg/mL of quinine sulphate in a form of the mean value versus arbitrary fluorescence units (FU). Therefore, the FIC data achieved is different from the general quantitative analysis by comparing the established calibration curve and is valuable for the qualitative analysis of the free fluorescence intermediary Maillard reaction products (MRPs) (10.1016/j.foodchem.2015.09.064.; 10.1016/j.foodchem.2005.01.027). To investigate the generation of the specific MRPs, chromatography and mass spectrometry analyses (LC-MS and GC-MS) are required.

  1. Results and discussion: L 174-175: “The degree of roasting can also be reflected by the alteration of the color parameters under certain circumstances” – In relation to potentially harmful Maillard reaction product it is so interesting, so please specify and add more details, and examples.

Changes: Thanks very much for the suggestions. Detailed information about the association between roasting degree and color parameters has been added. “In the case of barley malt, its roasting degree could be evaluated according to the development of high molecular weight browning compounds [28]. The association between roasting degree and color parameters had also been well established in coffee and hazelnuts [29,30]. Therefore, the color parameters of the RCHB were essential to be analyzed.” Please check Line 186-190 in the revised manuscript.

  1. Tables 1 and 3 - Please, add letters as a superscript.

Changes: Thanks very much for the suggestions. The letters in Table 1 and Table 3 had been modified to superscript accordingly. Please check the revised manuscript.

  1. L 207-210: “these results suggested that the color parameters of CHB roasted at different temperatures varied with the cultivars, in which the blue and black cultivars were less affected due to their relatively higher contents of anthocyanin and colored polyphenols in episperm initially” Please, describe this phenomenon briefly.

Changes: Thanks very much for the suggestions. The original conclusion had been modified to “these results suggested that the color parameters of the blue and black cultivars were less affected by the roasting temperature due to their relatively higher contents of anthocyanin and colored polyphenols.” Please check Line 222-224 in the revised manuscript.

  1. L264: “On the contrary, methylpyrazine and 2-methylbutanoic acid were negatively affected by roasting temperature” and L292-293: “Contrarily, the contents of aldehydes, ketones, acids, furans, and pyrazines were generally elevated after roasting and the impact of roasting temperature on their contents varied with the cultivars” L319-320: “As reported, the roasting temperature was a crucial factor that influenced the formations of pyrazines and aldehydes, presenting a positive and a negative correlation with their contents [36], separately” Can the author try to explain the reason for this exception?

Changes: Thanks very much for the suggestions.

(1) For the specific volatile compounds, the impact of roasting on their contents varied. For instance, the concentration of methylpyrazine in the roasted coffee was increased followed by a steep decrease with the elevated roasting temperature (10.1016/0963-9969(95)00037-2). As revealed in the manuscript, the relative content of acetic acid appeared to be promoted with roasting temperature (Figure 2a, 2d).

(2) Due to the diverse differences in the numbers and concentrations of individual compounds, a comprehensive assessment of the volatile substances on the basis of their chemical classes is more valuable for understanding the changes in aromas of CHB during the roasting treatment. In the present study, pyrazines were increased with the roasting temperature, suggesting to be the characteristic aroma indicators for the roasted CHB, whereas aldehydes appeared to be decreased with the elevated roasting temperature. It was reported that roasting-induced interaction between aldehydes and ketones was essential for the formation of pyrazines particularly under high roasting temperatures, indicating that aldehydes were crucial contributors to the aromas and participants in the formation of other volatiles (10.1002/ffj.3597). The analysis of pyrazines and aldehydes in the discussion has been updated. Please check Line 331-336 in the revised manuscript.

  1. L 303: “Nonetheless, it was still noteworthy to emphasize that the relative contents of furans and acids decreased with roasting temperature despite their elevations compared with those of the raw CHB.”

 It is worth describing this phenomenon in more detail.

Changes: Thanks very much for the suggestions. The analysis of changes in furans and acids has been updated as follows: “It was still noteworthy that the relative contents of furans and acids in the RCHB were negatively associated with their roasting temperature and the highest elevations in furans and acids among CHB roasted at 260.0 ℃ were observed in BK-260 and WH-260, revealing 43.16 % and 447.44 % higher contents than those of BK and WH, respectively. Under the same circumstances, the highest amounts of furans and acids appeared to be BU-260 (1.83 %) and BK-260 (4.80 %), separately, indicating that more acids were generated than furans, contrary to the findings in the roasted coffee bean [38].” Please check Line 315-322 in the revised manuscript.

  1. L315-318: “Pyrazines and aldehydes were usually demonstrated as the primary volatile substances of the roasted products (e.g. cocoa, soybean) and derived from the α-amino- ketones by the Strecker degradation, the Maillard reaction, and lipid oxidation, conferring nutty (furfural), earthy (trimethylpyrazine), cocoa (methylpyrazine), and roasted flavors (5-methylfurfural)” It should be mentioned earlier - > L294-303 (aldehydes and pyrazines). Moreover, Strecker degradation is one of the most important reactions leading to final aroma compounds in the Maillard reaction.

Changes: Thanks very much for the suggestions. “Pyrazines and aldehydes…” has been moved to the beginning of the paragraph as per your advice and the discussion associated with the Strecker degradation has been carefully checked and revised. Please check Line 325-329 in the revised manuscript.

  1. L382-386: “In contrast, the phenolic compounds of the blue cultivar were significantly decreased at the roasting temperature of 180.0 ℃ (p < 0.05), suggesting their sensitivity to thermal-induced oxidation and degradation [49]. Notably, the polyphenols contents in the white and blue cultivars were obviously promoted at the roasting temperature of 260.0 ℃, which might be associated with the formation of polyphenol derivatives (e.g. phenylindans) during the thermal processing [50].” It has been mentioned too briefly. More specific data and results should be presented (on the basis of the previous studies).

Changes: Thanks very much for the suggestions. The analysis of the polyphenols has been updated as follows: “In contrast, the phenolic compounds of the blue cultivar were significantly decreased at the roasting temperature of 180.0 ℃ (p < 0.05), suggesting their sensitivity to the thermal-induced decomposition (e.g. anthocyanin, cumaric acid, and quercetin), in accordance with the previous findings [35,51,52]. Notably, the polyphenols contents in the white and blue cultivars were obviously promoted at the roasting temperature of 260.0 ℃, which might be associated with the formation of polyphenol derivatives such as the degradation of caffeic acid induced the generation of phenylindans and the oxidation of polyphenols induced the generation of melanoidins during the thermal processing [53]. The MRPs might also contribute to the changes in their polyphenol contents [54].” Please check Line 402-410 in the revised manuscript.

  1. L387: “Besides, roasting seemed to have little impact on the polyphenols content of the black cultivar” It should be emphasized, whether there are statistically significant differences. However, the TPC method is not precise enough to find statistically significant differences in this case.

Changes: Thanks very much for the suggestions.

(1) The analysis of the polyphenols in the black cultivar has been updated as follows: “Besides, roasting had little impact on the polyphenols content of the black cultivar (p > 0.05).” Please check Line 410-411 in the revised manuscript.

(2) As mentioned in Q4, the determination of total polyphenols by the Folin-Ciocalteu method can be considered as supplementary evidence for the changes in color parameters and formations of the Maillard reaction products instead of the accurate quantitative analysis.

  1. L392: I suggest using “free intermediate compounds” instead advanced MRPs.

Changes: Thanks very much for the suggestions. The advanced MRPs have been modified to “free fluorescence intermediary compounds” accordingly. Please check the revised manuscript.

  1. Fig. 3 b - It would be easier and more clearly compare the results in a table.

Changes: Thanks very much for the suggestions. We have changed Figure 3b into Table 4 as per your advice. Please check the revised manuscript.

  1. L400: “It was suggested that the advanced MRPs were a heterogeneous group of compounds with characteristic pigmentation and their generation usually affected the color parameters.” Heat treatment during the roasting process affects the formation of potentially harmful Maillard reaction compounds, such as acrylamide, hydroxymethylfurfural, furfural, and especially advanced glycation end products (AGE) e.g., Nε-(carboxymethyl)lysine). In my opinion, determining the level of a well-known specific compound in the advanced Maillard reaction stage is a more specific and thus better solution than estimating the number of free intermediate compounds (FIC) characterized by fluorescence.

Changes: Thanks very much for the suggestions. We agree with you and accept your advice. Usually, roasting-induced formation of the Maillard reaction products is inevitable, particularly for the complex food matrix. Highland barley is an essential staple food for local Tibetans and is commonly processed with roasting, which safety should be paid more attention to. As mentioned above (Q5), chromatography and mass spectrometry analyses (LC-MS and GC-MS) are required to elucidate the qualitative and quantitative information of the MRPs. The assessments of the toxicity and biological property of the roasted CHB are in progress and the findings will be presented in our future manuscript. A short prospect “Nonetheless, the FIC values appeared to be the indicators for the MRPs and were unable to elucidate the specific compounds, which information still needed further chromatography and mass spectrometry analyses.” has been added. Please check Line 437-440 in the revised manuscript.

  1. L: 395- 397: “As expected, the roasting treatment of CHB resulted in an apparent increase in its FIC value whilst presenting a roasting temperature-dependent manner with the raise rates of 53.72 %~968.37 % (white), 12.63 %~718.32 % (blue)..). Please, discuss the results with other sources and add more information about FIC levels after heat treatment of other raw materials/ products.

Changes: Thanks very much for the suggestions. The analysis of FIC has been updated as follows:

“Consequently, the FIC values among CHB roasted at 260.0 ℃ were significantly different from each other (p < 0.05), in which WH-260 (1.15) showed the highest value followed by BU-260 (0.92). Roasting-induced promotions in FIC values had been observed in buckwheat and groats, where FIC values of the roasted buckwheat showed well relevance to the roasting time [14,56]. It was noteworthy that the FIC values of the RCHB were apparently lower compared with other referred products, which might be ascribed to the differences in the determination wavelength (excitation at 370.0 nm vs 353.0 nm and emission at 440.0 nm vs 438.0 nm).” Please check Line 426-432 in the revised manuscript.

  1. L 423: “whereas flavonoids were also negatively relevant to pyrazines, ketones, and acids (p < 0.05).” According to previous data, some phenolic compounds suppressed Maillard-type volatiles and carcinogenic product - acrylamide (doi.org/10.1016/j.foodchem.2019.125008; doi.org/10.1021/tx9001644). Please take this aspect into account in the discussion.

Changes: Thanks very much for the suggestions. The association between flavonoids and volatile substances has been updated as follows:

“Polyphenols and flavonoids showed opposite associations with esters (negative and positive, respectively), whereas flavonoids also showed negative relevance to pyrazines, ketones, and acids (p < 0.05), consistent with the previous reports, which demonstrated that several phenol acids (e.g. gallic acid and caffeic acid) and naringenin might involve in suppressing the formation of the Maillard-type products [57,58].” Please check Line 453-457 in the revised manuscript.

Round 2

Reviewer 2 Report

Please, check typing errors, e.g. Millard, etc.

Author Response

  1. Please, check typing errors, e.g. Millard, etc.

Changes: Thanks very much for the suggestions. The error of “Millard” has been revised. Please check Line 419 in the revised manuscript. Other errors have been carefully checked and revised as per your advice.